# How Self-Stigma Fuels Negative Coping Strategies During COVID-19: Behavioral Pathways Through Negative Emotions and Motivational Impairment

**DOI:** 10.3390/bs15101380

**Published:** 2025-10-11

**Authors:** Yifeng Wang, Kan Shi, Shuhui Xu

**Affiliations:** 1Laboratory of Ecological Civilization and Environmental Management, Wenzhou University, Wenzhou 325035, China; fs_wyf1997@foxmail.com (Y.W.); shik@psych.ac.cn (K.S.); 2Academy of Wenzhou Model Development Research, Wenzhou University, Wenzhou 325035, China; 3Department of Psychology, Wenzhou University, Wenzhou 325035, China; 4Institute of Higher Education, Wenzhou University, Wenzhou 325035, China

**Keywords:** self-stigma, mediation effect, negative emotions, inspirational motivation, COVID-19 pandemic

## Abstract

From a social epidemiology perspective, this study examines self-stigma among COVID-19 quarantine populations and its influence on negative coping strategies. An online survey of 292 residents from quarantine and non-quarantine zones assessed self-stigma, negative emotions, inspirational motivation, and coping behaviors. Results showed that quarantined individuals experienced higher negative emotions and lower inspirational motivation than non-quarantined individuals. Self-stigma was positively linked to negative emotions and maladaptive coping, and negatively linked to inspirational motivation. Mediation analysis revealed that negative emotions and inspirational motivation partially explained the effect of self-stigma on negative coping strategies. These findings highlight self-stigma as a significant social determinant affecting emotional and behavioral responses during quarantine. The study emphasizes the importance of integrating stigma assessment into mental health monitoring and suggests implementing stigma-reduction interventions to enhance psychological resilience in pandemic settings.

## 1. Introduction

### 1.1. Self-Stigma and Negative Coping Strategies

Individuals or groups who possess certain socially undesirable or disreputable traits, which lower their status in society, become stigmatized and experience a reduced self-perceived value. Stigma refers to the degrading and insulting labels society assigns to these individuals or groups ([6]). [11] ([11]) formally introduced the concept of “stigma” in Stigma: Notes on the Management of Spoiled Identity. Stigma research mainly refers to the reactions of the general public toward stigmatized groups. Self-stigmatization is the response of stigmatized individuals or groups, directing the stigmatizing attitudes inwardly ([33]). Public stigmatization is an important source of self-stigmatization, and when individuals internalize the stigmatizing views of the public, it leads to reduced self-esteem, impaired self-efficacy, and behaviors such as withdrawal, including social avoidance ([2]). Research shows that stigmatization and perceived threats significantly predict psychological distress and behavioral vigilance ([6]). Individuals who are stigmatized often experience persistent anxiety and depressive states ([39]) and may exhibit antisocial behaviors due to negative inducement ([10]). Self-stigma includes cognitive, emotional, and behavioral components and may lead to negative emotions such as fear, loneliness, anxiety, and depression, thereby negatively impacting mental health ([25]; [31]). This is especially true for individuals with depression-prone traits, as self-stigmatization reduces their self-esteem levels ([18]). Previous studies have focused on migrant populations, the LGBTQ+ community, and people with disabilities. However, research is limited on self-stigmatization in groups facing stigma during public health crises, such as individuals infected during the early COVID-19 quarantine.

From a theoretical standpoint, this study adopts a psychological perspective while explicitly acknowledging the broader epidemiological context. Although the primary mechanisms under investigation are cognitive and affective, individual-level responses to stigma are embedded within societal and public health processes. Integrating psychological insights with a population-level understanding allows us to examine coping strategies as both individual and socially consequential behaviors, thereby bridging micro- and macro-level analyses ([24]; [32]).

Coping strategies refer to cognitive and behavioral ways individuals manage difficulties and can be categorized as positive or negative. Negative coping strategies involve emotional tactics such as avoidance and denial, which individuals use to reduce discomfort from frustration or stress ([47]). Individuals adopting negative coping tend to engage in self-defensive behaviors, often relying on the alleviation of negative emotions like feelings of deprivation ([30]). Empirical studies have confirmed the impact of negative coping: for example, among high school students, negative coping predicted guilt ([19]); among medical students, negative coping correlated positively with mental health problems ([4]) and mediated the relationship between psychological capital and academic burnout ([26]) or workplace bullying and quiet quitting ([9]). Studies on perceived discrimination have also linked it to negative coping among immigrants in Spain ([8]) and Latino HIV-positive individuals ([1]). Taken together, self-stigmatization may influence coping choices in stressful contexts through similar mechanisms.

Therefore, this study seeks to examine cognitive and affective mechanisms through which self-stigma motivates the use of negative coping strategies, focusing on individuals infected during the early stages of quarantine.

### 1.2. The Mediating Role of Negative Emotions

Self-stigmatization may induce negative emotions, which in turn promote negative coping strategies. Self-stigmatization is closely related to negative emotional experiences ([36]; [21]), and risk events like the COVID-19 pandemic can also trigger negative emotions. Research shows negative emotions predict negative coping strategies ([13]), exacerbate rumination ([37]), impair cognitive function ([17]), and influence social decision-making ([44]).

Taken together, these findings suggest that negative emotions may serve as one pathway linking self-stigmatization to negative coping strategies. From an epistemic perspective, modeling negative emotions as mediators enables the integration of individual psychological processes with population-level outcomes. By considering how internalized stigma translates into affective responses that may influence behavioral patterns during public health crises, the study situates psychological mechanisms within an epidemiologically relevant framework ([7]). In the present study, negative emotions are modeled as a parallel mediator, operating simultaneously with inspirational motivation rather than sequentially.

### 1.3. The Mediating Role of Inspirational Motivation

External support factors, such as inspirational motivation from organizational or community leaders, may buffer the negative effects of self-stigmatization. Inspirational motivation refers to behaviors that enhance intrinsic motivation by clarifying common goals and future prospects ([16]). Research indicates inspirational motivation improves mental health ([43]), organizational commitment ([35]), and positive expectations during crises ([5]), while reducing negative outcomes such as turnover intentions ([3]). Given that self-stigmatization is associated with lower treatment-seeking, higher discontinuation, and worse symptoms ([20]), inspirational motivation may serve as an independent pathway mitigating the impact of self-stigmatization on negative coping.

This conceptualization highlights the interplay between individual psychological processes and socio-structural supports, reflecting a multi-level epistemic approach that integrates psychological theory with applied public health interventions. It emphasizes how leadership and community-level motivational strategies can buffer the negative consequences of self-stigma during epidemiological crises.

### 1.4. Overview and Hypotheses

In this study, we investigate how self-stigmatization among individuals in quarantine during the COVID-19 pandemic predicts negative coping strategies, with two potential mediators: negative emotions and inspirational motivation. Although sequential mediation can be considered in some contexts, the present study adopts a parallel mediation model, in which negative emotions and inspirational motivation are treated as simultaneous mediators with two independent indirect pathways. This approach is appropriate because (1) theoretical and empirical evidence suggests that negative emotions and inspirational motivation operate independently rather than in a fixed causal sequence ([36]; [16]; [43]), and (2) parallel mediation allows clearer estimation of the unique indirect effects of each mediator while controlling for the other, avoiding over-interpretation of causal order that is not supported by data.

The mediators are estimated using Hayes’ PROCESS macro (version 3.3) with Model 4, which implements a nonparametric Bootstrap procedure (5000 resamples) to compute bias-corrected confidence intervals for each indirect effect. This method is widely recommended for testing mediation because it does not assume normality of the indirect effect distribution and provides robust estimation of effect sizes ([15]; [34]).

Based on the literature, we propose the following hypotheses:

**H1:** *During the outbreak of the pandemic*, *self-stigmatization among individuals in quarantine areas significantly predicts their negative coping strategies*.

**H2:** *Negative emotions mediate the relationship between self-stigmatization and negative coping strategies among individuals in quarantine areas*.

**H3:** *Inspirational motivation mediates the relationship between self-stigmatization and negative coping strategies among individuals in quarantine areas*.

Figure 1 illustrates the conceptual parallel mediation model tested in this study, with self-stigmatization as the independent variable, negative coping strategies as the dependent variable, and negative emotions and inspirational motivation as simultaneous mediators.

## 2. Materials and Methods

### 2.1. Participants

The participants in this survey consisted of 147 individuals living in high-incidence epidemic areas (including both isolated and non-isolated zones) and those within infectious disease hospitals. Participants were residents of mainland China, recruited from multiple provinces and municipalities, including Hubei, Zhejiang, Xinjiang, Shanghai, Shandong, Guangdong, Jiangxi, Shaanxi, Jiangsu, Hebei, Beijing, Henan, Gansu, and Anhui. Among the participants, 70 were male, accounting for 47.6% of the sample, and 77 were female, accounting for 52.4%. The participants were categorized into age groups: 31 individuals were under 20 years old, accounting for 21.1% of the sample; 26 were aged 20–29, accounting for 17.7%; 41 were aged 30–39, accounting for 27.9%; 29 were aged 40–49, accounting for 19.7%; 15 were aged 50–59, accounting for 10.2%; and 5 were over 60 years old, accounting for 3.4%. In terms of education, 5 participants had middle school or lower education, accounting for 3.4%; 18 had a high school level of education (including vocational and technical schools), accounting for 12.2%; 17 had a college diploma, accounting for 11.6%; 55 had a bachelor’s degree, accounting for 35.4%; and 52 held a master’s degree or higher, also accounting for 35.4%.

For comparing variables between individuals in the quarantine area and those in non-quarantine areas, 145 samples were randomly selected from the non-quarantine zone. Of these, 55 were male, accounting for 37.9%, and 90 were female, accounting for 62.1%. The participants were divided into age groups: 27 were under 20 years old, accounting for 18.6%; 44 were aged 20–29, accounting for 30.3%; 34 were aged 30–39, accounting for 23.4%; 26 were aged 40–49, accounting for 17.9%; 13 were aged 50–59, accounting for 9%; and 1 was over 60 years old, accounting for 0.7%. In terms of education, 15 participants had middle school or lower education, accounting for 10.3%; 29 had a high school level of education (including vocational and technical schools), accounting for 20%; 33 had a college diploma, accounting for 22.8%; 55 had a bachelor’s degree, accounting for 37.9%; and 13 held a master’s degree or higher, accounting for 9%. The non-quarantine zone participants were randomly selected from a pool of 1071 individuals for comparative analysis with those in quarantine zones.

### 2.2. Measures

#### 2.2.1. Stigmatization Scale

The Stigmatization Scale was adapted from the Devaluation–Discrimination Scale (PDDS) developed by [23] ([23]) to measure public stigmatization in the context of COVID-19. The scale consists of 12 items and uses a five-point Likert scale ranging from 1 (“Strongly disagree”) to 5 (“Strongly agree”), with higher scores indicating greater perceived public stigmatization. A sample item is: “Most people don’t mind making close friends with those who have had COVID-19.” Items that are positively worded (i.e., agreement reflects lower stigmatization) were reverse-scored prior to analysis. Composite scores were calculated by summing all 12 items, yielding possible total scores ranging from 12 to 60. The Cronbach’s α coefficient for the 12 items in this study was 0.93.

#### 2.2.2. Negative Emotions Scale

The Negative Emotions Scale (DASS-21) ([28]) consists of 21 items, including three dimensions: depression, anxiety, and stress, with seven items for each dimension. A four-point Likert scale is used, ranging from 1 (“Does not apply”) to 4 (“Always applies”), with higher scores indicating more severe negative emotions. One sample item is: “I found it hard to wind down.” Composite scores were calculated by summing all 21 items, yielding possible scores ranging from 21 to 84. The Cronbach’s α coefficient for the 21 items in this study was 0.97.

#### 2.2.3. Negative Coping Strategies Scale

The Negative Coping Strategies Scale is based on the negative behavior subscale of the Coping Strategies Scale developed by [38] ([38]). The scale consists of three items and uses a five-point Likert scale ranging from 1 (“Never”) to 5 (“Always”), with higher scores indicating more negative coping strategies. One sample item is: “Tried to see the positive side of the situation (positive); let my feelings out somehow (negative).” Composite scores were calculated by summing the three items, yielding possible scores ranging from 3 to 15. The Cronbach’s α coefficient for the three items in this study was 0.72.

#### 2.2.4. Inspirational Motivation Subscale

The Inspirational Motivation Subscale is based on the vision motivation dimension of the Transformational Leadership Scale developed by [22] ([22]). The scale consists of six items, using a five-point Likert scale ranging from 1 (“Strongly Disagree”) to 5 (“Strongly Agree”), with higher scores indicating higher levels of inspirational motivation. One sample item is: “Paints a desirable future for everyone.” Composite scores were calculated by summing all six items, yielding possible scores ranging from 6 to 30. The Cronbach’s α coefficient for the six items in this study was 0.97.

### 2.3. Procedure

The survey was conducted nationwide through an online method. The questionnaire was designed and distributed via the Wen-Juan-Xing platform and disseminated to participants through social media platforms such as WeChat. In addition to open recruitment via social media, we coordinated with local community/neighborhood leaders and quarantine-area coordinators to notify residents about the study and facilitate participation. The instructions emphasized the confidentiality and authenticity of the responses. All participants provided informed consent prior to data collection. For individuals who did not regularly use social media or had difficulty completing the questionnaire online, trained personnel (e.g., community staff or designated survey assistants) provided guidance and assistance to complete the survey on-site or by phone, while still ensuring that informed consent procedures were followed. The research procedures complied with the ethical standards of the institution and the Declaration of Helsinki.

SPSS 21.0 software was used for reliability analysis, descriptive statistics, independent samples t-tests, and correlation analysis, while Hayes’ PROCESS macro (version 3.3) was employed to conduct the mediation effect testing. This macro applies a nonparametric Bootstrap resampling procedure (5000 iterations) to estimate the confidence intervals of indirect effects, which provides a robust test of mediation without relying on the normality assumption ([15]; [34]). Harman’s single-factor test was conducted for data analysis, which revealed that there were six factors with eigenvalues greater than 1 before rotation. The variance explained by the first factor was 37.54%, which is below the 40% threshold. This suggests that the sample does not exhibit significant common method bias ([46]).

## 3. Results

### 3.1. Descriptive Statistics and Correlation Analysis of Variables

Correlation analysis was conducted on the total scores of the four variables: self-stigmatization, negative emotions, negative coping strategies, and transformational leadership’s inspirational motivation. The results indicated that self-stigmatization was significantly positively correlated with negative emotions and negative coping strategies, and significantly negatively correlated with inspirational motivation. Negative emotions were significantly positively correlated with negative coping strategies but were not significantly correlated with inspirational motivation. Negative coping strategies were significantly negatively correlated with inspirational motivation (see Table 1 for details).

### 3.2. Comparison of Differences in Variables Between Individuals in Quarantine Areas and Non-Quarantine Areas

A comparison of the categorical differences in the four variables revealed that individuals in quarantine areas had significantly higher levels of negative emotions compared to those in non-quarantine areas. Additionally, the inspirational motivation scores of individuals in quarantine areas were significantly lower than those of individuals in non-quarantine areas. No significant differences were found for the other two variables (Table 2). The Bar Chart of *t*-test Results is shown in Figure 2.

### 3.3. Differences in Gender, Age, and Educational Attainment Among Participants in the Quarantine Area

Analyses of demographic variation revealed differential patterns across the four focal dimensions. Independent-samples t-tests indicated no significant gender differences between male (*N* = 70) and female (*N* = 77) participants on perceived stigma (*t*(145) = 0.60, *p* > 0.05, *d* = 0.10), negative affect (*t*(133.96) = −0.99, *p* > 0.05, *d* = −0.16), approach motivation (*t*(145) = −0.24, *p* > 0.05, *d* = −0.04), or avoidance motivation (*t*(134.71) = −1.86, *p* > 0.05, *d* = −0.31). One-way ANOVA across age groups showed a significant effect only for negative affect (*F*(5,141) = 2.565, *p* = 0.03); perceived stigma (*p* = 0.421), approach motivation (*p* = 0.662), and avoidance motivation (*p* = 0.096) did not differ significantly by age. With respect to educational attainment, ANOVA results revealed significant group differences for negative affect (*F* = 2.943, *p* = 0.023) and approach motivation (*F* = 5.044, *p* = 0.001), whereas perceived stigma (*p* = 0.346) and avoidance motivation (*p* = 0.093) showed no significant variation across education levels. Overall, demographic effects were most pronounced for negative affect and approach motivation, while perceived stigma and avoidance motivation remained largely invariant across the examined demographic strata.

### 3.4. The Relationship Between Self-Stigmatization and Negative Coping Strategies: Mediation Effect Testing

Controlling for gender, age, and educational attainment, we tested whether negative emotions mediate the relationship between self-stigmatization and negative coping strategies. Self-stigmatization significantly and positively predicted negative coping strategies (*β* = 0.41, *t* = 5.36, *p* < 0.001) and also significantly and positively predicted negative emotions (*β* = 0.44, *t* = 5.87, *p* < 0.001). In turn, negative emotions significantly and positively predicted negative coping strategies (*β* = 0.37, *t* = 4.63, *p* < 0.001) (see Table 3).

Bootstrapped mediation analysis with 5000 resamples indicated a significant indirect effect of self-stigmatization on negative coping strategies via negative emotions (indirect effect = 0.163; 95% CI [0.056, 0.305]), as the confidence interval does not include zero; this indirect effect accounted for 39.78% of the total effect. At the same time, the direct effect of self-stigmatization on negative coping strategies remained significant (direct effect = 0.247; 95% BootCI [0.089, 0.406]), indicating that negative emotions partially mediate the relationship between self-stigmatization and negative coping strategies (see Table 4).

Controlling for gender, age, and educational attainment, we examined whether approach motivation mediates the relationship between self-stigmatization and negative coping strategies. Self-stigmatization significantly and positively predicted negative coping strategies (*β* = 0.41, *t* = 5.36, *p* < 0.001), and it was a significant negative predictor of approach motivation (*β* = −0.22, *t* = −2.74, *p* < 0.01). The path from approach motivation to negative coping strategies did not reach statistical significance (*β* = −0.15, *t* = −1.95, *p* > 0.05). See Table 5 for path coefficients.

Bootstrapped mediation analysis (5000 resamples) yielded a small but statistically significant indirect effect through approach motivation (indirect effect = 0.034; 95% CI [0.002, 0.090]), indicating a reliable mediation despite the non-significant direct path; this indirect effect accounted for 7.32% of the total effect. Concurrently, the direct effect of self-stigmatization on negative coping strategies remained significant (direct effect = 0.377; 95% CI [0.223, 0.531]) and represented 92.68% of the total effect. These findings indicate that approach motivation partially mediates the association between self-stigmatization and negative coping strategies, although the mediating contribution is modest. See Table 6.

## 4. Discussion

First, individuals located in Quarantine Areas exhibited markedly different psychological profiles compared with those outside such zones. Independent-samples t-tests demonstrated that residents of Quarantine Areas reported significantly greater negative affect and significantly lower inspirational (approach) motivation. These findings are consistent with evidence that the COVID-19 pandemic engendered widespread psychological strain globally ([41]; [42]). Quarantine Areas, as geographically and socially delineated high-risk contexts, not only heighten objective threats to health but also concentrate negative information and social scrutiny, thereby intensifying negative emotional responses. Concurrently, the sustained demands of pandemic management appear to deplete psychological resources and diminish future-oriented optimism (i.e., inspirational motivation). Together, these patterns underscore the pronounced mental-health burden borne by residents of Quarantine Areas and the need for timely, targeted psychological support.

Second, self-stigmatization exerted a robust direct effect on negative coping strategies among residents of Quarantine Areas. In this study, self-stigmatization denotes the internalization of socially transmitted devaluation and discriminatory attitudes associated with COVID-19-related stereotypes, producing experiences of exclusion and adverse psychological consequences. As members of territorially marked high-risk groups, residents of Quarantine Areas are vulnerable to external stigmatizing processes that facilitate internalization. Our results show that self-stigmatization directly increases reliance on maladaptive coping (e.g., avoidance, self-blame), in line with prior empirical work on health-related stigma ([27]). The stigmatization sequence—labeling, differential treatment, rejection, and societal exclusion—often culminates in individuals adopting negative self-views, reduced self-esteem, and diminished self-efficacy, which in turn promote passive acceptance and maladaptive responses to stress ([40]). This mechanism provides a parsimonious account of the direct pathway from internalized stigma to negative coping strategies.

Third, mediation analyses indicate that negative emotions and inspirational motivation operate as parallel mediators linking self-stigmatization to negative coping strategies, but they differ substantially in effect magnitude. Specifically, the pathway via exacerbated negative emotions accounted for the bulk of the indirect effect (≈39.25%), whereas the pathway via weakened inspirational motivation accounted for a comparatively modest share (≈7.83%). The dominant role of negative emotions is consistent with stigma internalization models and modified labeling theory, which posit that external demeaning evaluations become incorporated into the self-concept and precipitate pervasive negative affect ([29]). Internalized stigma undermines self-worth and fosters an attentional bias toward threat and loss-related information, thereby amplifying anxiety, depressive symptoms, and other forms of negative affect ([14]). Sustained negative affect also disrupts prospective cognition and goal-directed planning ([12]), which reduces adaptive problem-solving and increases the likelihood of resorting to negative coping strategies when confronted with stressors.

Although the mediating contribution of inspirational motivation was smaller, it is theoretically meaningful. In our framework, inspirational motivation indexes an individual’s intrinsic, future-oriented drive and sense of purposeful expectation. Self-stigmatization undermines self-esteem and perceived competence, thereby eroding hope, agency, and goal pursuit. When individuals’ intrinsic motivational resources are weakened, they experience reduced perceived control and diminished expectation of successful outcomes, which lowers proactive coping and increases proneness to avoidance or resignation. This interpretation accords with psychological empowerment theory, which links perceived control and motivational resources to reduced passive coping ([45]). Thus, while negative affect constitutes the principal psychological conduit through which self-stigmatization translates into maladaptive coping, loss of inspirational motivation represents an additional, though modest, pathway that further compromises individuals’ capacity to engage adaptively with stressors.

Collectively, these findings refine our understanding of how internalized stigma operates in high-risk, quarantine contexts: self-stigmatization directly fosters maladaptive coping and also acts indirectly—predominantly by heightening negative affect and, to a lesser extent, by diminishing inspirational motivation. The differential effect sizes suggest that interventions aimed at ameliorating negative affect (for example, emotion-regulation training and targeted cognitive-behavioral approaches) are likely to yield the largest reductions in maladaptive coping, whereas strategies that bolster goal-directed motivation and psychological empowerment may provide complementary benefits.

## 5. Practical Implications

The following practical implications are grounded in data collected during the COVID-19 pandemic and therefore are most directly applicable to similar large-scale public-health emergencies or periods of elevated social stress; their generalization to routine, non-pandemic contexts should be tested in future research.

Framed through a social epidemiology lens, these findings highlight the necessity of integrating self-stigma into public health surveillance as a measurable social determinant: routine stigma assessments should be combined with epidemiological data to identify communities at elevated psychosocial risk, enabling targeted allocation of mental health resources and rapid deployment of support services where stigma—and its emotional sequelae—is most concentrated. In pandemic or outbreak settings, such targeted surveillance can help prioritize scarce mental-health personnel and digital support tools to areas experiencing acute stigma-related distress.

Large-scale, community-engaged anti-stigma campaigns are essential to disrupt quarantine-related stereotypes and foster social cohesion: partnering with local leaders and civil society organizations to co-create culturally tailored messages via mass media and social media channels can reduce “othering” of quarantined populations and attenuate the social isolation that exacerbates negative coping behaviors. We note that the effectiveness and uptake of such campaigns may vary outside of outbreak conditions; therefore, implementation in non-pandemic contexts should be accompanied by rigorous evaluation.

Emotional health monitoring must be embedded within quarantine protocols through validated brief scales administered via telehealth or digital check-in platforms: aggregating these data at the population level allows for real-time detection of distress clusters and triggers mobile mental health teams or digital mood support modules, ensuring timely intervention to prevent escalation of maladaptive coping. Pilot testing of these monitoring systems in routine community health settings is recommended before broad non-emergency rollout.

Finally, inspirational motivation interventions—delivered en masse via SMS, community Wi-Fi portals, or peer support testimonials—should be scaled and systematically evaluated for uptake and effectiveness across neighborhoods, while pandemic preparedness plans and funding frameworks explicitly recognize self-stigma and its downstream effects as key social determinants of health, allocating proportional resources to stigma-reduction programs in areas exhibiting the greatest need. We recommend that future studies empirically evaluate whether these intervention strategies retain effectiveness in non-pandemic environments and across diverse cultural and socioeconomic contexts.

## 6. Limitations and Future Research

This study also has some limitations that should be addressed in future research. On one hand, since this study solely relied on self-reported questionnaires, and all three variables in this research pertain to negative psychological behaviors, there is a possibility of social desirability bias in the responses, which may lead to false data. Current statistical methods are not yet effective in identifying and isolating such biases. Therefore, future research could incorporate behavioral experiments or case interviews to reduce the impact of these biases on the study’s conclusions.

On the other hand, the participants in this study were individuals from quarantine areas, such as those in infectious disease hospitals or high-risk epidemic zones, and the sample size is not large. As a result, it is not sufficient to reveal demographic characteristics effectively. Future studies could consider using longitudinal research designs to explore the temporal effects of self-stigmatization on negative coping strategies among individuals in quarantine areas and the underlying mechanisms. Although we tried to include participants with limited social-media use via community staff and trained assistants, recruitment was mainly online and may over-represent people with better digital access or stronger community ties. This potential selection bias limits generalizability. Future studies should use mixed-mode or probability-based sampling, record social-media use as a covariate, apply weighting to correct sampling imbalances, and replicate findings in offline or non-pandemic settings.

## Figures and Tables

**Figure 1 behavsci-15-01380-f001:**
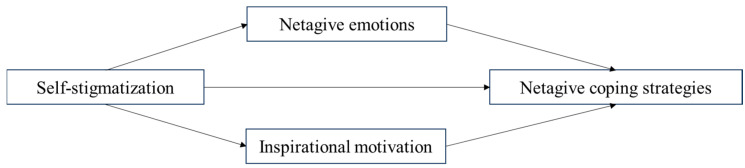
Hypothetical Model.

**Figure 2 behavsci-15-01380-f002:**
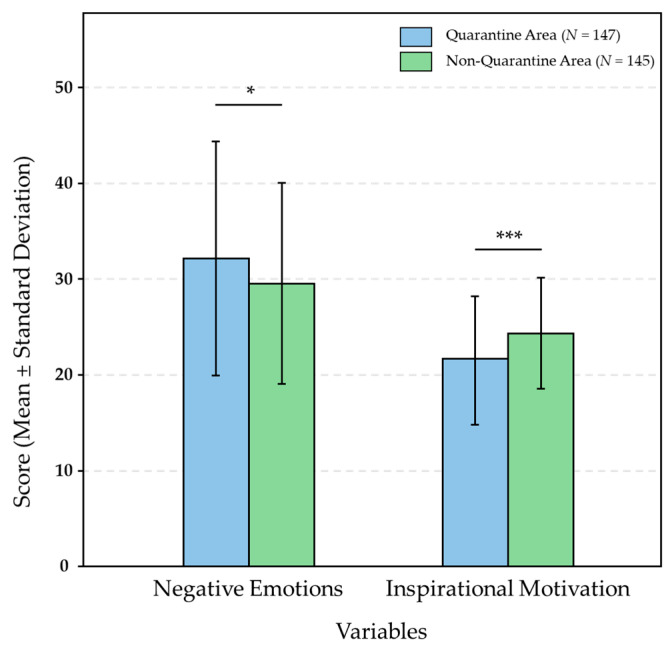
Comparison of Psychological Variable Scores Between Quarantine and Non-Quarantine Areas. Note: * *p* < 0.05, *** *p* < 0.001.

**Table 1 behavsci-15-01380-t001:** Descriptive Statistics and Correlation Matrix of Variables.

Variables	*M*	*SD*	1	2	3	4
1. Self-Stigmatization	30.24	10.55	1			
2. Negative Emotions	32.26	12.19	0.42 ***	1		
3. Negative Coping Strategies	5.27	2.82	0.40 ***	0.47 ***	1	
4. Inspirational Motivation	21.68	6.17	−0.23 **	−0.10	−0.22 **	1

Note: ** *p* < 0.01, *** *p* < 0.001.

**Table 2 behavsci-15-01380-t002:** Independent Samples t-test of Variable Scores Between Individuals in Quarantine and Non-Quarantine Areas.

Variables	Category		*t*	Cohen’s *d*
	Quarantine Area (*N* = 147)	Non-Quarantine Area (*N* = 145)		
Self-Stigmatization	30.24 ± 10.55	28.27 ± 9.31	1.69	0.20
Negative Emotions	32.27 ± 12.19	29.64 ± 10.48	1.97 *	0.23
Negative Coping Strategies	5.27 ± 2.82	4.93 ± 2.79	1.02	0.12
Inspirational Motivation	21.68 ± 6.71	24.43 ± 5.75	3.76 ***	0.44

Note: * *p* < 0.05, *** *p* < 0.001.

**Table 3 behavsci-15-01380-t003:** Mediating Effect of Negative Emotions between Self-Stigmatization and Negative Coping Strategies (*N* = 147).

Variables	Negative Emotions			Negative Coping Strategies			Negative Coping Strategies		
	*β*	*SE*	*t*	*β*	*SE*	*t*	*β*	*SE*	*t*
Self-Stigmatization	0.44	0.08	5.87 ***	0.25	0.08	3.10 **	0.41	0.08	5.36 ***
Negative Emotions				0.37	0.08	4.63 ***			
Gender	−0.04	0.16	−0.26	0.18	0.15	1.20	0.17	0.16	1.02
Age	−0.12	0.06	−1.97	−0.06	0.06	−0.98	−0.11	0.06	−1.64
Level of education	−0.02	0.08	−0.32	0.11	0.07	1.49	0.10	0.08	1.27
*R* ^2^	0.21			0.29			0.18		
*F*	9.23 ***			11.53 ***			7.92 ***		

Note: ** *p* < 0.01, *** *p* < 0.001.

**Table 4 behavsci-15-01380-t004:** Mediation Effect Test of Self-Stigmatization on Negative Coping Strategies through Negative Emotions.

Effect Type	Effect	BootSE	BootLLCI	BootULCI	Effect Proportion(%)
Indirect Effect	0.163	0.063	0.056	0.305	39.78%
Direct Effect	0.247	0.080	0.089	0.406	60.22%
Total Effect	0.411	0.077	0.259	0.562	

**Table 5 behavsci-15-01380-t005:** Mediating Effect of Inspirational Motivation between Self-Stigmatization and Negative Coping Strategies (*N* = 147).

Variables	Inspirational Motivation			Negative Coping Strategies			Negative Coping Strategies		
	*β*	*SE*	*t*	*β*	*SE*	*t*	*β*	*SE*	*t*
Self-Stigmatization	−0.22	0.08	−2.74 **	0.38	0.08	4.83 ***	0.41	0.08	5.36 ***
Inspirational Motivation				−0.15	0.08	−1.95			
Gender	0.32	0.17	1.86	0.21	0.16	1.32	0.17	0.16	1.02
Age	0.02	0.07	0.35	−0.10	0.06	−1.60	−0.11	0.06	−1.64
Level of education	0.02	0.08	0.28	0.10	0.08	1.33	0.10	0.08	1.27
*R* ^2^	0.07			0.20			0.18		
*F*	2.81 *			7.22 ***			7.92 ***		

Note: * *p* < 0.05, ** *p* < 0.01, *** *p* < 0.001.

**Table 6 behavsci-15-01380-t006:** Mediation Effect Test of Self-Stigmatization on Negative Coping Strategies through Inspirational Motivation.

Effect Type	Effect	BootSE	BootLLCI	BootULCI	Effect Proportion(%)
Indirect Effect	0.034	0.023	0.002	0.090	7.32%
Direct Effect	0.377	0.077	0.223	0.531	92.68%
Total Effect	0.411	0.077	0.259	0.562	

## Data Availability

The data are available on request from the corresponding author. The data are not publicly available due to privacy or ethical restrictions.

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
