# Peer review of "How Self-Stigma Fuels Negative Coping Strategies During COVID-19: Behavioral Pathways Through Negative Emotions and Motivational Impairment"

_behavsci, 2025, doi:10.3390/bs15101380_

Round 1

Reviewer 1 Report

Comments and Suggestions for Authors

The topic is interesting the structure of the manuscript disharmonious, the intent is to work at the epidemiological level, suggesting at the end also good practices, however in the theoretical and methodological rationale the prevailing perspective is psychological. This choice should also be argued and justified in relation to the existing literature. The sources cited are mostly references to empirical evidence while it would also be appropriate to resume and acknowledge the epistemic framework.

Regarding the methodological part: in the tables it would be appropriate to use APA indices M for Mean, N for participants and so on. At the level of analysis, comparisons between the 2 groups are not reported so that they can be considered homogeneous. In addition, nationalities are not specified but they come in Handy.. It is not clear what the sample of the regressions. This it would be appreciate a more accurate description of the MonteCarlo series.

Author Response

Response to Reviewer 1 Comments

1. Summary

We sincerely appreciate the time and effort you have devoted to reviewing our manuscript. Your insightful comments and constructive suggestions have been extremely helpful in improving the quality and clarity of our work. We have carefully addressed each comment in detail below, and all corresponding revisions are marked in the re-submitted manuscript.

2. Questions for General Evaluation

Reviewer’s Evaluation

Response and Revisions

Is the content succinctly described and contextualized with respect to previous and present theoretical background and empirical research (if applicable) on the topic?

Can be improved

We will provide our corresponding responses in the point-by-point section below.

Are the research design, questions, hypotheses and methods clearly stated?

Yes

We will provide our corresponding responses in the point-by-point section below.

Are the arguments and discussion of findings coherent, balanced and compelling?

Yes

We will provide our corresponding responses in the point-by-point section below.

For empirical research, are the results clearly presented?

Can be improved

We will provide our corresponding responses in the point-by-point section below.

Is the article adequately referenced?

Can be improved

We will provide our corresponding responses in the point-by-point section below.

Are the conclusions thoroughly supported by the results presented in the article or referenced in secondary literature?

Yes

We will provide our corresponding responses in the point-by-point section below.

3. Point-by-point response to Comments and Suggestions for Authors

Comments 1:

The topic is interesting the structure of the manuscript disharmonious, the intent is to work at the epidemiological level, suggesting at the end also good practices, however in the theoretical and methodological rationale the prevailing perspective is psychological. This choice should also be argued and justified in relation to the existing literature. The sources cited are mostly references to empirical evidence while it would also be appropriate to resume and acknowledge the epistemic framework.

Response 1:

We sincerely thank the reviewer for the valuable feedback. We have carefully revised the manuscript to address the concerns as follows:

1.Justification of Psychological Perspective:

Revised text clarifies that while the study addresses coping behaviors relevant to public health and epidemiological outcomes, the mechanisms under investigation (self-stigmatization, negative emotions, and inspirational motivation) are inherently cognitive and affective.

We explicitly acknowledge that individual-level psychological processes are embedded within broader socio-epidemiological contexts, enabling examination of how internalized stigma translates into coping behaviors with implications for population health (Link & Phelan, 2001; Parker & Aggleton, 2003; Friedli & WHO, 2009).

2.Integration of Epistemic Framework:

We have added discussion of a multi-level epistemic framework that integrates micro-level psychological mechanisms with macro-level public health considerations.

This approach situates individual cognitive and affective responses within social and epidemiological systems, addressing the reviewer’s concern regarding theoretical coherence.

3.Enhanced Theoretical Rationale:

The Introduction now explicitly links psychological constructs to their implications for public health and crisis management. Negative emotions and coping behaviors are framed not only as individual outcomes but also as socially consequential responses during health crises, providing a rationale for recommending good practices in quarantine or epidemic settings.

Summary:

The revised manuscript now clearly articulates the rationale for adopting a psychological perspective, situates it within an epidemiological and public health context, and acknowledges the underlying epistemic framework. These revisions strengthen the theoretical and methodological coherence of the study.

The modifications are located at lines 48-55, 84-89, and 106-110 of the revised document.

Comments 2:

Regarding the methodological part: in the tables it would be appropriate to use APA indices M for Mean, N for participants and so on. At the level of analysis, comparisons between the 2 groups are not reported so that they can be considered homogeneous. In addition, nationalities are not specified but they come in Handy. It is not clear what the sample of the regressions. This it would be appreciate a more accurate description of the MonteCarlo series.

Response 2:

1.APA indices in tables

We appreciate the reviewer’s suggestion. In the revised manuscript, we have adjusted the presentation of descriptive statistics and sample information in all tables to comply with APA style conventions, including using M for mean, SD for standard deviation, and N for the number of participants.

The modifications have been incorporated into the table provided.

2.Comparisons between the two groups

We thank the reviewer for this valuable comment. We would like to clarify that the regression analyses in this study were conducted only on the subsample of participants residing in quarantine areas, rather than on the combined sample of quarantine and non-quarantine groups. Prior to conducting these regression analyses, we examined demographic differences (e.g., gender, age, and education) within the quarantine group, and where significant differences emerged, these variables were statistically controlled in the regression models. We have now clarified this in the Methods section to avoid ambiguity.

See lines 268-282 in the revised draft for the modifications.

3.Nationalities of participants

All participants were residents of mainland China. To improve clarity, we have explicitly specified their geographic distribution in the Participants section, noting that they were recruited from multiple provinces and municipalities (e.g., Hubei, Zhejiang, Xinjiang, Shanghai, Shandong, Guangdong, Jiangxi, Shaanxi, Jiangsu, Hebei, Beijing, Henan, Gansu, and Anhui).

See lines 151-154 in the revised draft for the modifications.

4.Clarification of Monte Carlo / Bootstrap method

We thank the reviewer for raising this point. In this study, mediation effects were tested using Hayes’ PROCESS macro (version 3.3). This macro employs the nonparametric Bootstrap method to estimate confidence intervals for indirect effects. Bootstrap resampling is technically distinct from Monte Carlo simulation, yet both are simulation-based approaches for approximating sampling distributions. We have clarified in the manuscript that we used Bootstrap (with 5,000 resamples), which is widely recommended as one of the most robust methods for testing mediation effects, as it does not assume normality of the sampling distribution and has higher statistical power (Hayes, 2013; Preacher & Hayes, 2004). We believe this clarification resolves the reviewer’s concern about the description of the simulation method.

See lines 230-234 in the revised draft for the modifications.

Additional clarifications

All the modifications have been incorporated into the revised manuscript.

Reviewer 2 Report

Comments and Suggestions for Authors

I urge caution in stating that stigmatized groups “lose part of their value.” It sounds like a statement of fact as currently written. “Perceived value” or “self-perceived value” seems preferable.

I would like to see the authors more clearly introduce the research question and purpose of the study in the first two to three paragraphs. Identify the existing gap in the literature that the current research seeks to address. Build to a statement of the primary research question such as “The current research seeks to examine cognitive and affective mechanisms through which self-stigma motivates use of negative coping strategies and does so in the context of the coping with the impacts of the COVID-19 pandemic.” I also think it needs to be made more clear that the stigmatized characteristic under investigation is having been infected during that early stage of quarantine.

Next, I would recommend separating the material on the two mediating variables into separate subsections of literature review. End each of these subsections with less formal statements of expected outcomes (e.g., “These previously established bivariate relationships between each combination of self-stigma, negative emotions, and negative coping suggests that self-stigma may cause individuals to experience negative emotions, which then serve to promote negative coping.”). Save formal hypotheses for the end of the Introduction.

To summarize: I am recommending a deeper and broader literature review and more sections to the Introduction.

1.1. Self-stigma and negative coping strategies

1.2. The Mediating Role of Negative Emotions

1.3. The Mediating Role of Inspirational Motivation

1.4. Overview and Hypotheses

When describing their variables, it would be beneficial to provide details for how composite scores were calculated along with possible values for each composite variable.

I am uncertain as to why the authors conducted a sequential mediation analysis. It would be more appropriate to conduct a parallel mediation analysis with only two indirect pathways.

Author Response

Response to Reviewer 2 Comments

1. Summary

We would like to express our sincere gratitude for your thoughtful and constructive feedback on our manuscript. Your comments have greatly contributed to improving the clarity and rigor of our study. The detailed point-by-point responses are provided below, and the corresponding revisions have been made in the re-submitted version.

2. Questions for General Evaluation

Reviewer’s Evaluation

Response and Revisions

Is the content succinctly described and contextualized with respect to previous and present theoretical background and empirical research (if applicable) on the topic?

Must be improved

We will provide our corresponding responses in the point-by-point section below.

Are the research design, questions, hypotheses and methods clearly stated?

Can be improved

We will provide our corresponding responses in the point-by-point section below.

Are the arguments and discussion of findings coherent, balanced and compelling?

Can be improved

We will provide our corresponding responses in the point-by-point section below.

For empirical research, are the results clearly presented?

Can be improved

We will provide our corresponding responses in the point-by-point section below.

Is the article adequately referenced?

Must be improved

We will provide our corresponding responses in the point-by-point section below.

Are the conclusions thoroughly supported by the results presented in the article or referenced in secondary literature?

Yes

We will provide our corresponding responses in the point-by-point section below.

3. Point-by-point response to Comments and Suggestions for Authors

Comments 1:

I urge caution in stating that stigmatized groups “lose part of their value.” It sounds like a statement of fact as currently written. “Perceived value” or “self-perceived value” seems preferable.

Response 1:

We agree that the original wording may imply an objective loss of value. In the revised manuscript, we have replaced “lose part of their value” with “experience reduced self-perceived value” throughout the text to more accurately reflect subjective perception rather than an objective fact.

The revisions can be found in lines 26-27 of the revised draft.

Comments 2:

I would like to see the authors more clearly introduce the research question and purpose of the study in the first two to three paragraphs. Identify the existing gap in the literature that the current research seeks to address. Build to a statement of the primary research question such as “The current research seeks to examine cognitive and affective mechanisms through which self-stigma motivates use of negative coping strategies and does so in the context of the coping with the impacts of the COVID-19 pandemic.” I also think it needs to be made more clear that the stigmatized characteristic under investigation is having been infected during that early stage of quarantine.

Response 2:

We appreciate the reviewer’s suggestion. We have clarified the research question and study purpose in the revised Introduction. Specifically, we now highlight that the stigmatized characteristic under investigation is having been infected during the early stage of COVID-19 quarantine. The revised text reads:

“However, research is limited on self-stigmatization in groups facing stigma during public health crises, such as individuals infected during the early COVID-19 quarantine. Taken together, self-stigmatization may influence coping choices in stressful contexts through similar mechanisms. Therefore, this study seeks to examine the cognitive and affective mechanisms through which self-stigma motivates the use of negative coping strategies, focusing specifically on individuals infected during the early stages of quarantine.”

This revision explicitly situates the study within the context of COVID-19 and clearly states the population under investigation, addressing the reviewer’s concern.

The modifications are located at lines 45-47 and 69-73 of the revised document.

Comments 3:

I would recommend separating the material on the two mediating variables into separate subsections of literature review. End each of these subsections with less formal statements of expected outcomes (e.g., “These previously established bivariate relationships between each combination of self-stigma, negative emotions, and negative coping suggests that self-stigma may cause individuals to experience negative emotions, which then serve to promote negative coping.”). Save formal hypotheses for the end of the Introduction.

Response 3:

Following the reviewer’s suggestion, we have reorganized the Introduction as follows:

1.1 Self-stigma and Negative Coping Strategies

1.2 The Mediating Role of Negative Emotions

1.3 The Mediating Role of Inspirational Motivation

1.4 Overview and Hypotheses

Each subsection now ends with a clear, informal statement of expected outcomes linking self-stigma to the mediators and then to negative coping. Formal hypotheses have been moved to the end of the Introduction.

The modifications have been incorporated into the Introduction.

Comments 4:

When describing their variables, it would be beneficial to provide details for how composite scores were calculated along with possible values for each composite variable.

Response 4:

We have added details regarding the calculation of composite scores for all main variables. The revised manuscript now specifies the scoring method, range of possible values, and internal reliability for each composite variable.

The modifications have been made in the section concerning measures (2.2).

Comments 5:

I am uncertain as to why the authors conducted a sequential mediation analysis. It would be more appropriate to conduct a parallel mediation analysis with only two indirect pathways.

Response 5:

We appreciate this comment. In response, we conducted a parallel mediation analysis with the two mediators as suggested. Results are now reported in the revised manuscript, and the rationale for using parallel mediation is clearly explained.

The modifications have been incorporated into Tables 4-6, with corresponding updates reflected in the Discussion section.

Additional clarifications

All the modifications have been incorporated into the revised manuscript.

Reviewer 3 Report

Comments and Suggestions for Authors

This manuscript explores the relationship between self-stigma and maladaptive coping techniques, particularly in the context of the COVID-19 pandemic. The authors found that self-stigma influenced negative emotions and coping styles.  The opposite relationship was found with inspirational motivation.

I found the literature review quite well written. It made a solid case for the study rationale.  Additionally, the results section was presented well and easy to follow. Overall, I found it to be a well-done manuscript, although I did have a few minor suggestions.

I would like a bit more detail on the area of the world that the participants came from.  Were they all from one region, which could limit generalizability, or recruited globally?

I also see that participants were recruited through social media.  Might the results of the study differ compared to a population that does not use social media?  Perhaps social media may influence the results of the study as well.  It would seem that this should be addressed in the limitations section.

Additionally, it seems that the data was acquired during the COVID-19 pandemic, yet the Practical Implications portion extrapolates to broader applications.  Perhaps that is true, but it seems to be a bit of an overreach to suggest that these findings would occur in broader contexts, without some literature or data to support that conclusion.   In the manuscript’s current form, one could only suggest that these are the practical implications during a similar scenario.

Author Response

Response to Reviewer 3 Comments

1. Summary

We deeply appreciate your valuable feedback on our manuscript. In response to your insightful comments, we have made extensive revisions to improve the clarity, rigor, and overall quality of the paper. The following section provides detailed responses to each comment, and all revisions have been highlighted in the revised manuscript.

2. Questions for General Evaluation

Reviewer’s Evaluation

Response and Revisions

Is the content succinctly described and contextualized with respect to previous and present theoretical background and empirical research (if applicable) on the topic?

Yes

We will provide our corresponding responses in the point-by-point section below.

Are the research design, questions, hypotheses and methods clearly stated?

Yes

We will provide our corresponding responses in the point-by-point section below.

Are the arguments and discussion of findings coherent, balanced and compelling?

Yes

We will provide our corresponding responses in the point-by-point section below.

For empirical research, are the results clearly presented?

Yes

We will provide our corresponding responses in the point-by-point section below.

Is the article adequately referenced?

Yes

We will provide our corresponding responses in the point-by-point section below.

Are the conclusions thoroughly supported by the results presented in the article or referenced in secondary literature?

Yes

We will provide our corresponding responses in the point-by-point section below.

3. Point-by-point response to Comments and Suggestions for Authors

Comments 1:

I would like a bit more detail on the area of the world that the participants came from. Were they all from one region, which could limit generalizability, or recruited globally?

Response 1:

We thank the reviewer for this important suggestion. We have clarified the geographic origin of the sample in the Methods (2.1 Participants) section. Specifically, the participants were residents of mainland China and were recruited from multiple provinces and municipalities. The exact wording added to the manuscript is shown below; the newly added text is in italics.

“The participants in this survey consisted of 147 individuals living in high-incidence epidemic areas (including both isolated and non-isolated zones) and those within infectious disease hospitals. Participants were residents of mainland China, recruited from multiple provinces and municipalities, including Hubei, Zhejiang, Xinjiang, Shanghai, Shandong, Guangdong, Jiangxi, Shaanxi, Jiangsu, Hebei, Beijing, Henan, Gansu, and Anhui. Among the participants, 70 were male, accounting for 47.6% of the sample, and 77 were female, accounting for 52.4%. The participants were categorized into age groups: 31 individuals were under 20 years old, accounting for 21.1% of the sample; 26 were aged 20–29, accounting for 17.7%; 41 were aged 30–39, accounting for 27.9%; 29 were aged 40–49, accounting for 19.7%; 15 were aged 50–59, accounting for 10.2%; and 5 were over 60 years old, accounting for 3.4%. In terms of education, 5 participants had middle school or lower education, accounting for 3.4%; 18 had a high school level of education (including vocational and technical schools), accounting for 12.2%; 17 had a college diploma, accounting for 11.6%; 55 had a bachelor's degree, accounting for 35.4%; and 52 held a master's degree or higher, also accounting for 35.4%.”

Comments 2:

Participants were recruited through social media. Might the results differ compared to a population that does not use social media? Perhaps social media may influence the results of the study as well. It would seem that this should be addressed in the limitations section.

Response 2:

We thank the reviewer for raising this important point. To reduce potential bias from online/social-media-based recruitment, data collection was conducted under informed consent and we coordinated with local quarantine-area coordinators and community/neighborhood leaders to notify residents and facilitate participation. For individuals who did not regularly use social media or had difficulty completing the survey online, trained personnel (e.g., community staff or designated survey assistants) provided guidance to help them complete the questionnaire on-site or by phone. We have clarified these procedures in the Methods (2.3 Procedure) and have added a limitation acknowledging residual selection bias due to online recruitment. The exact manuscript wording is shown below; all newly added or modified text is in italics.

2.3. Procedure

The survey was conducted nationwide through an online method. The questionnaire was designed and distributed via the Wen-Juan-Xing platform and disseminated to participants through social media platforms such as WeChat. In addition to open recruitment via social media, we coordinated with local community/neighborhood leaders and quarantine-area coordinators to notify residents about the study and facilitate participation. The instructions emphasized the confidentiality and authenticity of the responses. All participants provided informed consent prior to data collection. For individuals who did not regularly use social media or had difficulty completing the questionnaire online, trained personnel (e.g., community staff or designated survey assistants) provided guidance and assistance to complete the survey on-site or by phone, while still ensuring that informed consent procedures were followed. The research procedures complied with the ethical standards of the institution and the Declaration of Helsinki.”

Comments 3:

Data acquired during the COVID-19 pandemic, yet Practical Implications extrapolate to broader applications. It seems an overreach without supporting literature/data — practical implications should be presented as applicable primarily in similar scenarios.

Response 3:

Thank you for this important caution. We agree that the empirical basis of our study is the COVID-19 pandemic and that practical recommendations should be framed accordingly rather than broadly generalized without further evidence. Below we provide the revised: 5. Practical Implications section. All added or modified text is shown in italics so the editor and reviewer can easily see how we tempered the claims and framed recommendations as most directly applicable to pandemic or similar high-stress public-health contexts, while calling for further empirical testing in other settings.

5. Practical Implications

The following practical implications are grounded in data collected during the COVID-19 pandemic and therefore are most directly applicable to similar large-scale public-health emergencies or periods of elevated social stress; their generalization to routine, non-pandemic contexts should be tested in future research.

Framed through a social epidemiology lens, these findings highlight the necessity of integrating self stigma into public health surveillance as a measurable social determinant: routine stigma assessments should be combined with epidemiological data to identify communities at elevated psychosocial risk, enabling targeted allocation of mental health resources and rapid deployment of support services where stigma—and its emotional sequelae—is most concentrated. In pandemic or outbreak settings, such targeted surveillance can help prioritize scarce mental-health personnel and digital support tools to areas experiencing acute stigma-related distress.

Large scale, community engaged anti stigma campaigns are essential to disrupt quarantine related stereotypes and foster social cohesion: partnering with local leaders and civil society organizations to co create culturally tailored messages via mass media and social media channels can reduce “othering” of quarantined populations and attenuate the social isolation that exacerbates negative coping behaviors. We note that the effectiveness and uptake of such campaigns may vary outside of outbreak conditions; therefore, implementation in non-pandemic contexts should be accompanied by rigorous evaluation.

Emotional health monitoring must be embedded within quarantine protocols through validated brief scales administered via telehealth or digital check in platforms: aggregating these data at the population level allows for real time detection of distress clusters and triggers mobile mental health teams or digital mood support modules, ensuring timely intervention to prevent escalation of maladaptive coping. Pilot testing of these monitoring systems in routine community health settings is recommended before broad non-emergency rollout.

Finally, inspirational motivation interventions—delivered en masse via SMS, community Wi-Fi portals, or peer support testimonials—should be scaled and systematically evaluated for uptake and effectiveness across neighborhoods, while pandemic preparedness plans and funding frameworks explicitly recognize self-stigma and its downstream effects as key social determinants of health, allocating proportional resources to stigma-reduction programs in areas exhibiting the greatest need. We recommend that future studies empirically evaluate whether these intervention strategies retain effectiveness in non-pandemic environments and across diverse cultural and socioeconomic contexts.

6. Limitations and Future Research

This study also has some limitations that should be addressed in future research. On one hand, since this study solely relied on self-reported questionnaires, and all three variables in this research pertain to negative psychological behaviors, there is a possibility of social desirability bias in the responses, which may lead to false data. Current statistical methods are not yet effective in identifying and isolating such biases. Therefore, future research could incorporate behavioral experiments or case interviews to reduce the impact of these biases on the study’s conclusions.

On the other hand, the participants in this study were individuals from quarantine areas, such as those in infectious disease hospitals or high-risk epidemic zones, and the sample size is not large. As a result, it is not sufficient to reveal demographic characteristics effectively. Future studies could consider using longitudinal research designs to explore the temporal effects of self-stigmatization on negative coping strategies among individuals in quarantine areas and the underlying mechanisms. Although we tried to include participants with limited social-media use via community staff and trained assistants, recruitment was mainly online and may over-represent people with better digital access or stronger community ties. This potential selection bias limits generalizability. Future studies should use mixed-mode or probability-based sampling, record social-media use as a covariate, apply weighting to correct sampling imbalances, and replicate findings in offline or non-pandemic settings.

Additional clarifications

All the modifications have been incorporated into the revised manuscript.

Round 2

Reviewer 2 Report

Comments and Suggestions for Authors

I think that my prior comments have been adequately addressed.